# Efficient and Secured Mechanisms for Data Link in IoT WSNs: A Literature Review

**Muhammad Zulkifl Hasan** and **Zurina Mohd Hanapi** *

Department of Communication Technology and Networking, Universiti Putra Malaysia,
Seri Kembangan 43400, Selangor, Malaysia
* Correspondence: zurinamh@upm.edu.my

**Abstract:** The Internet of things (IoT) and wireless sensor networks (WSNs) have been rapidly and tremendously developing recently as computing technologies have brought about a significant revolution. Their applications and implementations can be found all around us, either individually or collaboratively. WSN plays a leading role in developing the general flexibility of industrial resources in terms of increasing productivity in the IoT. The critical principle of the IoT is to make existing businesses sufficiently intelligent to recognize the need for significant fault mitigation and short-cycle adaptation to improve effectiveness and financial profits. This article presents efficiently applied security protocols at the data link layer for WSN and IoT-based frameworks. It outlines the importance of WSN–IoT applications as well as the architecture of WSN in the IoT. Our primary aim is to highlight the research issues and limitations of WSNs related to the IoT. The fundamental goal of this work is to emphasize a suggested architecture linked to WSN–IoT to enhance energy and power consumption, mobility, information transmission, QoS, and security, as well as to present practical solutions to data link layer difficulties for the future using machine learning. Moreover, we present data link layer protocol issues, attacks, limitations, and research gaps for WSN frameworks based on the recent work conducted on the data link layer concerning WSN applications. Current significant issues and challenges pertain to flow control, quality of service (QoS), security, and performance. In the context of the literature, less work has been undertaken concerning the data link layer in WSN and its relation to improved network performance.

**Keywords:** wireless sensor networks (WSNs); Internet of things (IoT); computer network security; data link layer; IR4.0; information technology; machine learning

## 1. Introduction

The Internet of things (IoT) and wireless sensor network (WSNs) have seen a rapid and massive transformation in recent years, when all the computer science domains, operating independently or collaboratively, have seen unprecedented change, and their technologies and deployments can be observed all around us. A wireless sensor network (WSN) is a network that connects and collaborates. Its sensors are placed in different environments to collect the best data [1]. WSNs are made up of remote nodes that have a lot of promise for various businesses and are built on ad-libbed system architectures [2]. According to a report, a new network uprising has just recently begun, with approximately 50 billion items and smartphones expected to be connected to the Internet by 2020 [3]. The ever-increasing number of internet-related things is transforming the world we live in. Smart cities, network security management, e-health, traffic control, smart shopping, pollution control, radiation level detection, online education, cloud computing, intruder detection, smart parking, vehicle auto-fault diagnostics, and many other implementations of the IoT and WSN are only a few examples of this transformation [4]. The demographics of the WSN application spectrum palette are shown in Figure 1 [5].

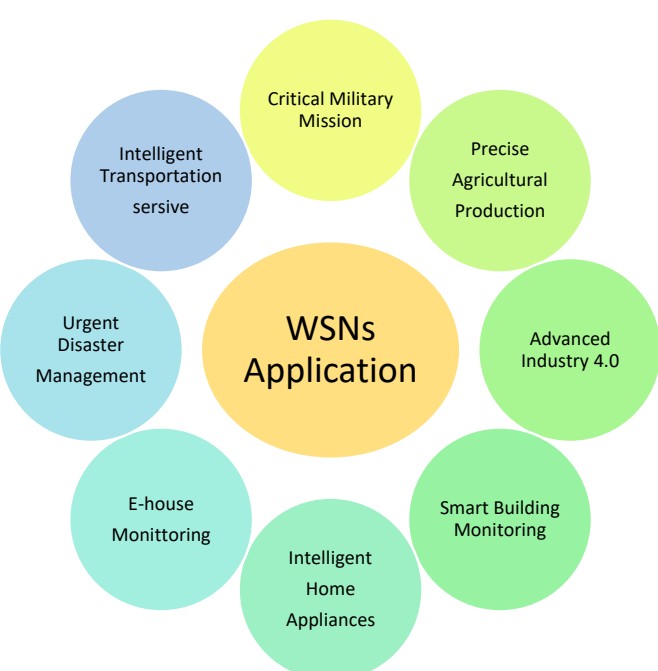

**Figure 1.** Application area spectrum of wireless sensor networks (WSNs) [5].

One of WSN's applications can be found in the clinical and medical sciences. Consequently, medical frameworks are being developed that collect health data from the human body using wearable sensors that can also be embedded within an individual's body. These sensors communicate the gathered information, which can be further monitored and processed to obtain various insights [6–8]. Figure 2 shows 2020's top IoT applications and the enterprise share of IoT projects worldwide [9]. The IoT serves as a backbone for big data and business analytics. A considerable amount of data can be retrieved, transferred, and further processed; this data can be utilized to tackle business issues, customer support, and service needs [10]. The utilization of AI strategies can create an effective decision support system. A.I. creates continuous prediction models that can be used. When the nodes attain the data point, they pass it to the mode [11].

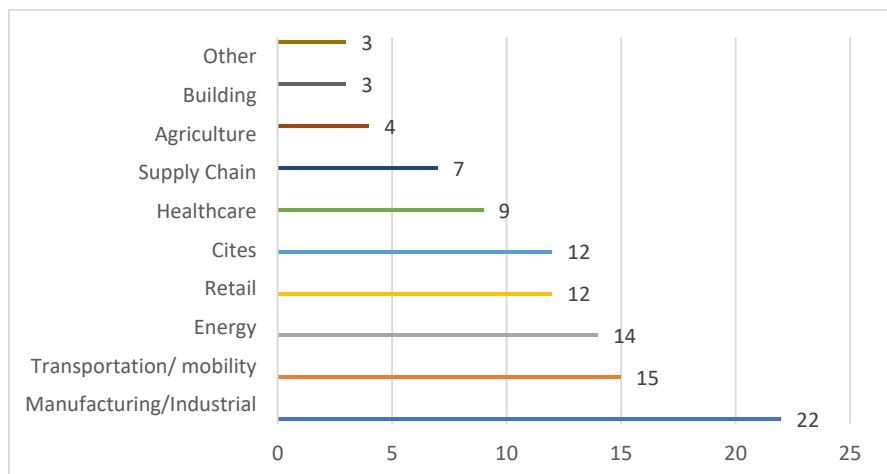

**Figure 2.** Top areas of IoT application in 2020 [12].

WSN makes use of a sensor network architecture. The OSI architectural model is the most common architecture for wireless sensor networks. WSN's structure comprises five primary and three cross layers. It is common practice in sensor n/w to use all five levels. In addition to hospitals and schools, roadways and buildings may also benefit from this

design, which can be used for several purposes, including disaster and crisis management and security. The two types of WSN architecture are layered network architecture (LNA) and clustered architecture (CA).

## 2. Layered Network Architecture

A base station and a high number of sensor nodes are used in this network. Network nodes may be arranged in concentric circles. The structure consists of three cross layers and five interlocking layers. The five architectural layers are as follows:

- Application layer;
- Transport layer;
- Network layer;
- Data link layer;
- Physical layer.

The following are examples of the three cross layers:

- Power management plane;
- Mobility management plane;
- Task management plane.

To improve overall network efficiency, these cross layers are primarily utilized to govern the network and make the sensors act as one. An architectural diagram of WSN as well as WSN's five layers is shown below in Figure 3.

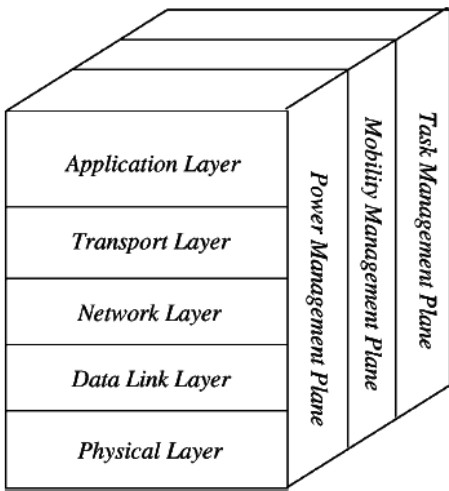

**Figure 3.** Architectural diagram of WSN (Reprinted from ref. [13]).

However, the main focus of this research is the data link layer, so a comprehensive discussion of a data link is included below:

1. **Data Link Layer**

The data connection layer is a program's protocol layer that regulates data transit through and out of the physical link of a network. The Open Systems Interconnection (OSI) model is used for telecommunication protocols [14]. Data bits are decoded, organized, and encoded at the data link layer until they are transferred as frames between two nearby nodes on the same LAN or WAN [15]. The data connection layer also manages the recovery of devices from collisions that occur when many nodes attempt to send frames simultaneously. The data link layer has two sub-layers: the logical connectivity control (LLC) sub-layer and the network access control (MAC) sub-layer [16]. The communication channel that links the adjacent nodes is known as the tie, and each datagram must be sent via a separate connection from the source to the destination [17].

The main role of the data connection layer is to transport datagrams across a single link. The data link layer protocol defines the packet structure and behavior for packets ex-

changed across nodes, including error detection, retransmission, flow control, and random access [18,19]. It allows nodes to use a traditional medium and make effective use of it to control data flow. It also handles transmission troubles. The most common attacks on the data link layer arise from the medium access control (MAC) sub-layer, collisions, and jamming [20]. The analysis of these assaults reveals new information about the timing considerations of MAC protocols in terms of security. In the absence of suitable counter-measures, analysis suggests which group of protocols protects against assaults [21]. The data link layer assures the stability of point-to-point (or point-to-multipoint) connections via multiplexing data frames, data streams, the MAC, and error control.

2. **Clustered Network Architecture**

This architecture relies on the "LEACH protocol", since it uses clusters to group sensor nodes. "LEACH protocol" is an acronym for "low-energy adaptive clustering hierarchy". Clustering is implemented in a two-tier structure. Sensor nodes are organized into clusters using this distributed approach. The TDMA (time-division multiple access) plans are created by the cluster head nodes in each individually formed cluster. The energy consumption of a network is reduced because of the data fusion concept. The ability to combine data in this network design makes it incredibly popular. In any cluster, all nodes may access data by interacting with the cluster head. The base station will receive data from all the clusters as mentioned in Figure 4. The process of forming a cluster, as well as the selection of its leader in each cluster, is both independent and autonomous [13].

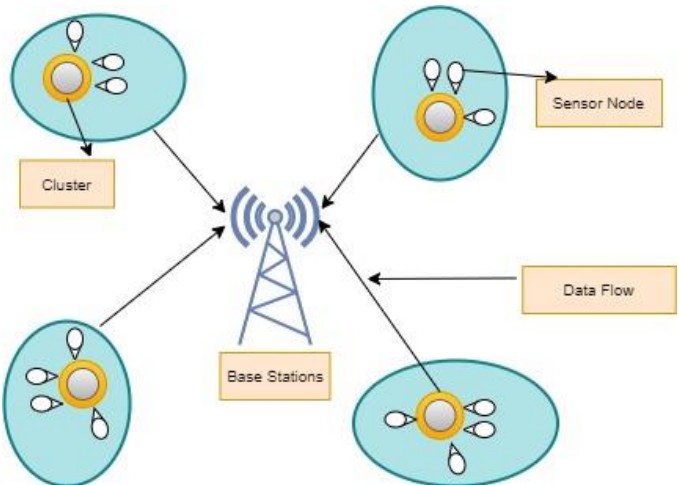

**Figure 4.** Clustered network architecture (Adapted from ref. [13]).

## 3. WSN in IOT

Heterogeneous WSNs that connect a diverse set of intelligent sensors have formed the foundation for IOT-based systems all around us, promising significant advancements shortly. With the rapid expansion of these technologies, there has been an increase in the temptation to reduce their energy use. Tremendous advances in communication and information flow have contributed to unsustainable increases in energy consumption and carbon emissions. However, sensor nodes must operate successfully for extended periods (even years) in most applications because of various application criteria (e.g., environmental management, agriculture, border surveillance or protection, etc.). Dead nodes may affect data reliability, precision, and device compatibility, which are essential for an application's long-term sustainability. A sensor node is typically composed of four primary units: the processing unit, the sensing/identification unit, the communication unit, and the power supply unit, as shown in Figure 5. Filters, amplifiers, transducers, comparators, and other secondary components are combined with the core above. Data from the workplace are collected and detected by the sensor device. All the other devices require power, which is supplied by the power unit (usually a battery-limited one) and

delivered to the BS (base stations) through the communication unit, which performs data processing functions, including data collection, as well as data manipulation duties, such as data gathering. The quantity of energy that a sensor node utilizes is based on its present state, which may be one of three states: sleeping, idle, or active. In active mode, the node uses the most significant energy. Due to the transmission and reception of information, the sensing device can release as much energy as is feasible while absorbing as little as possible. Though the energy the processing unit uses is far less than that required by the radio subsystem, it is more significant than that required by the sensor subsystem. There is a relationship between factors such as communication distance, monitoring cases, operational criteria, and the activities taken by each unit. When the node is idle, it waits for data packets to arrive from another node. Data transfer may waste 50% to 100% more energy if more power is required to run the CPU, radio, and other components. When the node is resting, substantially less energy is lost since no processing is undertaken and the communication unit is switched off. However, other energy dissipation problems exist, such as packet losses, packet collisions, physical channel challenges, frame overhearing, overhead protocols, and overhead processing. As a result, IoT researchers have been driven to develop energy-efficient and renewable IoT solutions [22]. Figure 5 shows the typical IoT architecture for WSN.

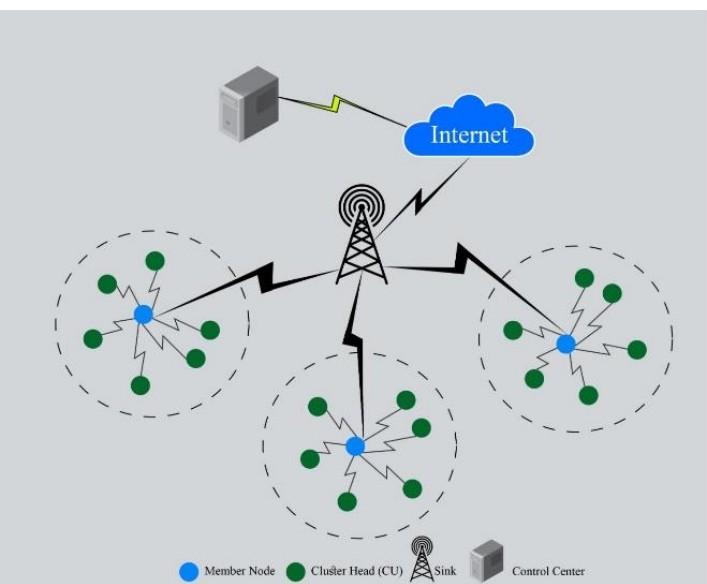

**Figure 5.** Typical IoT architecture for WSN.

Smart technology is becoming an increasingly important part of human life. WSNs are becoming more critical to the Internet of things every day. Consequently, the obstacles intensify, and the demand for reform becomes even more imperative. WSNs and the Internet of things may benefit from machine learning. A new operational framework for next-generation wireless sensor networks based on the IoT has been proposed. Using a three-layer transmission method, the nodes can communicate efficiently concerning energy.

However, there is a need to develop an IoT-based WSN architecture that is flexible, automated, and provides encryption in a lightweight manner. The proposed architecture also provides a reliable connection among the WSN's nodes based on privileges to maintain and exchange information. Similarly, for mobility, the proposed architecture uses dynamic clustering techniques. The cluster heads (CH) are rotated regularly in an iterated manner using a local dynamic cluster selection process with the primary goal of increasing the QoS and balancing their power consumption. Similarly, for encryption, the technique merely conducts rudimentary calculations by combining random permutations, replacements, and bit-level rotations. This suggested approach may dynamically modify the complexity of the encryption process depending on the currently available resources of each sensor

node. Furthermore, the technique enables dynamic essential creation and management with low processing among the WSN nodes. As a result, the suggested architecture would be highly resistant to assaults such as brute force, eavesdropping, man-in-the-middle, and replay attacks.

WSN is a vital component of IoT systems. The four critical difficulties discussed in this paper include high node density, node mobility, traffic heterogeneity, and the integration of WSNs with the cloud-based IoT. As part of the IoT-based Web of Things, this dissertation assesses how to connect WSNs to the Internet and proposes a vision for services such as heterogeneous traffic. Sensor deployment is a critical difficulty for WSNs that must be overcome. Using WSNs to identify and send information with a low delay is the emphasis of this effort. The integration of WSNs into the IoT and WSNs into the Internet are also discussed. We illustrate Internet of things (IoT) applications that use radio frequency identification (RFID) and wireless sensor networks (WSNs).

This study also demonstrates an IoT strategy for temperature monitoring in hospitals, aiming to combine and integrate low-cost and smart sensors across vast areas. In remote monitoring, the management layer, node layer, and cloud-based layer are critical components of IoT design. We present an in-depth look at each of these layers in detail [23].

## 4. Research Issues of WSN in IOT

The first difficulty for all uses of the WSNs is security. Protecting one's private and confidential information has become a top priority for modern consumers. Data regarding "personal health" and corporate operations, for example, should not be made public. To preserve their privacy and security, they must be transmitted above WSNs. Authentication and encryption are critical stages in safeguarding WSNs, but they alone are not adequate. Data security must be maintained as new security issues arise with the development of technology, such as the interconnection of the IoT and WSNs.

Despite recent developments in this sector, WSNs require substantial power from energy-restricted batteries to analyze and send data. Because of their limited size and computer capacity, wireless sensor nodes cannot perform to a great extent. WSNs have long been utilized in harsh, difficult-to-reach environments. Wireless sensor node resource limitations represent another issue for WSN-based systems.

Apps' potential to interact with sensors, other users, and the cloud is called "coverage and connectivity".

- Data aggregation methodologies;
- How to use sensors in a distributed environment;
- Clustering algorithms;
- Localization techniques;
- Rerouting protocols.

The differences between IoT devices raise the question of interoperability. The WSN or IoT environment must be able to interface with the many heterogeneous devices that generate various types of data. With the increasing variety of IoT applications and linked devices, a continued effort is needed to achieve this.

Facilitating communication among fast-moving objects, such as cars, and leveraging mobility to improve communication efficiency are difficult tasks in an age of constantly progressing Internet of things (IoT) systems. The mobility problem of WSNs and in the IoT sector has been addressed several times in the literature. Sensor nodes in the WSN scenario are only allowed to move inside the deployed zone of significance. As a result of the ubiquitous distribution of things in IoT scenarios, it is reasonable to expect that addressing the problem of things mobility would require the implementation of specific upgraded procedures.

The reuse of IoT devices is necessary because of the rising need for essential information (such as heart rate, temperature, blood pressure, etc.) in different WSN or IoT applications. The ability to use gadgets in a variety of ways saves money. It is therefore always a goal to design a device for an application that can be reused in future applications.

Things and sensor nodes already communicate regularly using current IoT and WSN systems. Battery life is quickly depleted by these exchanges, limiting non-stop operation to a few hours or days. As a result, improving processing and communication energy efficiency should be considered a significant open problem [24]. Many remedies have been presented in the literature to rectify this issue. To manage gathered data or events through different current solutions and services, the Internet of things (IoT) requires effective tactics. For IoT systems, the scalability criteria of WSNs cannot be met because of the higher number of IoT devices or other items linked to create an IoT system compared with conventional WSNs. Thus, the challenge of IoT scalability and flexibility must be addressed. An outstanding question rests in how flexible subscription and event monitoring systems may be provided while ensuring scalability for both objects and users.

Various applications have experienced delays due to multiple devices connected to sensors that provide a wide variety of data. Delay-sensitive IoT gadgets have proven common in WSN networks. Sensor-based IoT network latency has been investigated. Due to various factors, such as topology, radio interference, and mobility, this problem may emerge as the number of IoT data sources grows; it is necessary to consider this direction. WSN security is confined to a region interested in classical WSNs and IoT systems. They are seen as more secure because of the widespread use of internet-connected devices. To maintain a system secure from intrusions or attackers, it is critical to treat security and privacy as an open topic that requires ongoing improvement [25]. Many academics have presented security and privacy solutions, such as blockchain technology; however, continuing to overcome the numerous intimidations made by attackers remains a fascinating and inspiring challenge.

## 5. The Architecture of WSN Nodes

Wireless sensor network is a broad term that consists of several nodes; the more significant the WSN, the bigger the number of nodes. Each node acts as an individual unit consisting of a sensing unit, a communication unit, a processing unit, and a storage unit; these units make up a particular node [26]. The sensing unit exists to detect events and gather required data, such as temperature, humidity, sounds, light, etc., or the specific data it uses. The communication unit allows the collected data to be transferred; it makes the sensor communicate with other nodes for sending and receiving data. The storage unit is used to save the assembled data in a specific format for later use [27,28]. Figure 6 shows the architecture of an individual WSN node [29]. A WSN is made up of hundreds of thousands of sensor nodes. These nodes can communicate with one another using multi-hop communication. WSNs have a lot of potential as a platform for various uses, including data collection and event prediction.

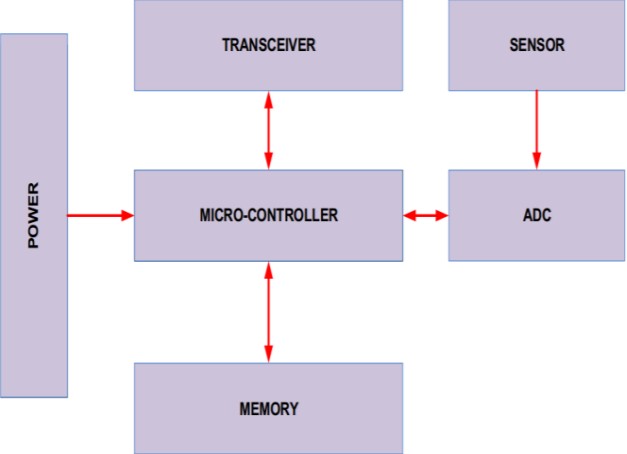

**Figure 6.** Architecture of an individual node [30].

## 6. IOT Architecture and Its Layers

This section describes the IoT infrastructure layers. These layers build up the IoT infrastructure, and the IoT frameworks would not seem to exist without them [31]. Each layer performs a vital task as well as process-specific operations. Figure 7 depicts the IoT layers from the bottom up (application layer, service layer, network layer, and physical layer) along with the responsibilities of each layer [32].

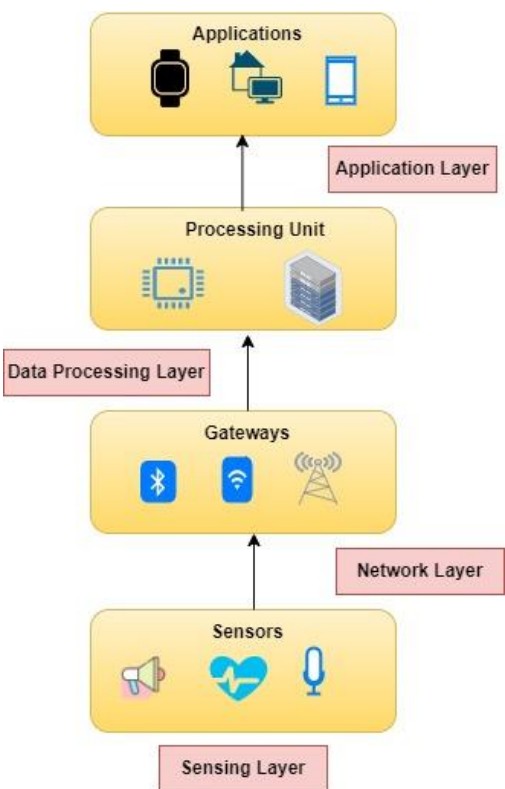

**Figure 7.** IoT architecture [32].

- **Application Layer**

The application layer characterizes all the applications where it is implemented. It serves as the link or medium between user-end devices and the IoT network. There are numerous IoT applications, and the application layer can aid the applications. Because of administrations based on sensor data, the administrations may be unique for each application. It is implemented by a dedicated program at the device end. The software uses the application layer on a computer [33] and supports application-layer protocols such as HTTPS, HTTP, FTP, and SMTP. The application layer uses a variety of protocols, including the restricted application protocol (CoAP), message queue telemetry transport (MQTT), the advanced message queuing protocol (AMQP), and an extensible messaging and presence protocol (XMPP) [34].

- **Data Processing Layer**

The service layer comprises functionalities that manage gathered data and link it to the component layer's data. This layer serves as a conduit between different IoT devices and provides advanced techniques for communicating between them [35]. The sensors are also connected to the application layer through the service layer, which sits on top of the network layer. It has two responsibilities: First, it verifies that information is submitted by legitimate clients while preventing all dangers and threats. Second, it demonstrates that information is sent by legitimate clients while avoiding all risks and threats [36].

- **Network Layer**

The Internet of things necessitates adaptability in operating a vast range of computers. Over a billion smartphones will be added to the framework every year. As a result, IPv6 will play a critical role in preserving network layer flexibility. This layer comprises network interchange programming as well as actual segments. Its fundamental object is to send information among gadgets and devices to the recipients [37,38]. The data is transmitted via the network layer, whether wired or wirelessly, using existing advanced methods.

- **Sensing Layer**

The sensing layer comprises the essential equipment, the gadget layer, hubs, and sensors, such as RFID, standardized tag names, actuators, and insightful identification gadgets [39]. Gadgets assemble and pass data to the network layer either directly or by implication. To discern the nodes, sensors are used to transport the gathered information into the next layer. It is predicted that all devices will be IPv6-competent in a few years' time [40,41].

## 7. Research Gap in WSN Layers

Table 1 presents the issues of WSN along with their limitations. The problems are mapped with their constraints, and gray areas are highlighted; we propose attendance to these current weaknesses in order to devise solutions. The table also includes recommendations and research development requirements in the context of WSN issues. A WSN node consists of various resources, such as power, memory, security, processing, network, and so on. Thus, according to the findings, significant problems are due to limited resources, and the required research developments reveal the need to build low-powered frameworks. Still, low power does not mean that security, efficiency, or reliability must be compromised. There is a demand for more reliable, trustworthy, efficient, and secure frameworks. Nevertheless, a low-powered operating system is necessary because WSN is deployed at locations where human intervention is not easily possible.

**Table 1.** WSN issues, limitations, and required research development.

| Sr No | Issue | Limitation | Required Research Development |
|-------|-------|------------|------------------------------|
| 1. | Limitation of resources [42–44] | Power Memory Internet Security | There is a need to develop low-powered, low memory, controlled security, and reliable transmission mediums for functional, secure, and fast data communication, which will also prolong the lifetime of the WSN. |
| 2. | Tampering [29,45,46] | Physical tampering Network tampering | Reliable network monitoring, network management, and network security protocols prevent or control physical and network attacks and ensure security. The WSN is impenetrable, and the network's performance is unaffected. |
| 3. | Security [47–49] | Network attacks Eavesdropping Access control Privacy | Researchers need to develop a solution to secure the network and ensure unapproved access, and the hub should maintain access control. The network needs to adapt to the various security conventions. |
| 4. | Operating system [50,51] | Complex systems Management Administration of resources | The OS of the sensor should be a less complex, simple programming environment that is fundamental to memory management. The application engineers should focus on validation activities such as planning, acquiring, and system administration. |
| 5. | Quality of service [52,53] | Network management Resource management | The QoS application's precise limits include organizational participation, dynamic sensors, efficiency, an estimate of sensor detail, dormancy, and delay accuracy. Further, the WSN's QoS can sustain extension, cancelling nodes. It is challenging to track QoS boundaries for sensor networks as the geography of the enterprise tends to evolve and data management is unclear. |

**Table 1.** *Cont.*

| Sr No | Issue | Limitation | Required Research Development |
|---|---|---|---|
| 6. | Deployment [54–56] | Node organization<br>Data security<br>Deployment strategy | Sensor hubs sent in such situations are incredibly dense; network blockage can occur due to multiple simultaneous transmission efforts. Sensor hubs configured in this case are insufficient or insufficiently numerous, resulting in a low information yield or a lack of data measurement. The property of self-setup is necessary if hubs are distributed randomly. |
| 7. | Robustness and fault tolerance [57,58] | Defective sensor<br>Sensor failure<br>Attack<br>Network failure | Sensor networks are vulnerable to power outages and faulty sensors as a result of any attack or change in environmental conditions. |
| 8. | Privacy and data confidentiality [59,60] | Spoofing<br>Unauthorized access<br>Modification of data<br>Sniffing | The IoT device should determine if the user or device has been authorized access to the system. Data access should be limited by granting or denying permission based on a set of rules. |
| 9. | Acquisition and transmission of data [60] | One-time-pad OTP encryption<br>Wireless transmission security | There is a need to develop a mechanism akin to OTP that monitors end-to-end communication links to identify vulnerabilities in applications and DBMS. |
| 10. | Authorization [61,62] | DNS attack<br>Biometric authentication<br>Defined role access | Passwords must be updated regularly, and computers should never be left unattended. Similarly, both the sender and the receiver need to perform authentication. |

## 8. WSN Security Protocols and Their Issues

Table 2 presents a list of implemented WSN security protocols and their purposes. The table states the number of specific network attacks that the WSN protocols are effective against. The attacks are characterized under three network performance parameters: authentication, confidentiality, and integrity. Each protocol can withstand certain attacks, which highlight the protocol's strength, while characterization helps to classify the damage executed by the attacks to the network performance parameters for the mentioned protocols. Moreover, the current significant security, performance, and usability issues that emphasize the weaknesses and vulnerabilities of each protocol are described.

**Table 2.** List of security protocols with their characteristics and issues.

| Protocol | Purpose | Effective against Attacks | Characteristics | Issues |
|---|---|---|---|---|
| TINYSEC [59] | To solve the inadequacy of existing systems, the TinySec architecture is used. TinySec is lightweight, reliable, and secure, ensuring message integrity, confidentiality, and access control. TinySec decreases energy consumption, bandwidth, and latency by more than 10%. | Gray hole, black hole, worm hole | Authentication Confidentiality | TinySec uses a single network-wide key such that any node in the network can impersonate any other node. It makes no effort to defend against various threats such as replay attacks, hello flood, node cloning, Sybil, path contamination, etc. |
| SPINS [63] | This is a series of protocols based on the use of two protected building units. The first is SNEP, which assures data security, authenticity, and richness. The second is TESLA, which enables authorized broadcasting in constrained environments. | Eavesdropping, gray hole, black hole, worm hole, replay, hello flood, node replication, Sybil, route poisoning | Authentication Confidentiality Integrity | The protocol states that if the nodes participating in the data are far from the source node and the nodes in between are disinterested in the data, the data will not be forwarded to the destination. As a result, it is not the best option for high-density node spread. |

**Table 2.** *Cont.*

| Protocol | Purpose | Effective against Attacks | Characteristics | Issues |
|---|---|---|---|---|
| MINISEC [64] | MiniSec is a stable and energy-aware protocol for WSNs. It uses fewer resources than other protocols such as TinySec and has the same amount of protection as ZigBee. OCB mode is used for authentication. This method saves time by allowing confidentiality and authentication to be granted in one step. | Eavesdropping, black hole, replay, flood, node poisoning | Authentication Confidentiality Integrity | MiniSec cannot be used to encrypt broadcast communications directly. If a node receives packets from many receiving nodes simultaneously, it must keep track of each sender's counter, which consumes a lot of memory. |
| LEAP [65] | LEAP is a security protocol that satisfies essential security needs such as anonymity and authentication. The critical exchange technique enables in-network processing while restricting a node violation's security impact on the breached node's immediate network neighborhood. | Eavesdropping, gray hole, black hole, replay, flood, Sybil | Authentication Confidentiality | LEAP's flaws are that it uses a tweaked protocol implementation that does not adequately secure the end user's credentials; thus, this information is vulnerable to being hacked. LEAP is only intended for wired networks and is not intended for use with untrusted wireless media. |
| ZIGBEE [66] | ZigBee's trust management model encourages other devices to access the network while still distributing keys. It consists of two categories of network entities: full-function devices that coordinate and reduced-function devices that are end devices. For low-security home applications, the Trust Center employs a residential mode. Commercial Mode is designed for business applications requiring a high security level. | Black hole, replay, route poisoning | Authentication Confidentiality Integrity | Short range is one of ZigBee's most significant drawbacks. High data speed and low complexity, high maintenance costs, a lack of a complete solution, and poor materialization are all factors to consider. Poor reception and network reliability are also drawbacks, putting ZigBee at a disadvantage compared with others. |
| LLSP [67] | LLSP uses the TinySec packet format as its basis. It is an energy-efficient connection-layer security protocol that guarantees message secrecy, authentication, message integrity checks, message security, and access control, among other things. It can refuse a request early and has allowed for performance overhead. | Replay, flood, route poisoning, gray hole, black hole, worm hole | Authentication Confidentiality Integrity | LLSP has the drawbacks of consuming more symbols and being more difficult to customize than others. The LLSP protocol provides periodic updates and a time-to-live value for the details, but it does so quickly, resulting in a weak liveliness indicator. |
| LISP [68] | LiSP is built on the concept of crucial renewability. To prevent keystream replay, it creates a new key each time. LiSP enables flexible any-to-any WAN access, encourages virtual machine mobility, increases scalability by aggregating more RLOCs, and supports simplifying multihomed routing. | Gray hole, black hole, worm hole, replay, route poisoning | Authentication Confidentiality Integrity Availability | Extra headers are added to LiSP packets, increasing the packet size while decreasing the payload available. Any modification to the mapping system is disseminated across the network due to the signaling process. This may cause packet loss or add latency to the system. Although LiSP specifies how to send various types of addresses in control messages, it does not specify how to execute look-up operations on any of these addresses. |
| LEDs [69] | The static and location-aware aspects of sensor networks suggest an adaptive security architecture to achieve end-to-end security. LEDs are a protocol that is based on position. This protocol's primary management system incorporates location-aware data, an additional detail that improves the protocol's resistance to key compromise and node capture attacks. | Eavesdropping, black hole, flood, node replication, replay, and route poisoning are all examples of cyberattacks. | Authentication Confidentiality Integrity | The LEDs protocol can only function if the network setup is predictable; otherwise, it will be unsuccessful due to a lack of complicated routing support. |

### 9. Network Attacks in WSNs

WSNs are subject to a variety of risks; attackers can interfere with radio transmissions by inserting their data bits into the link and replaying old packets, among other methods [40]. Attackers may put malicious nodes in the network with similar capabilities to normal nodes or catch typically deployed nodes and erase their memory [62]. Table 3 depicts a wide range of network security attacks mapped against network layers: the physical layer, data link layer, network layer, transport layer, and application layer. These network security attacks are categorized as routing-based and protocol-layer-based attacks. The type of network attack triggered by an attacker on any layer depends upon the layer type. These network layers consist of specific functions and restricted responsibilities; thus, an attack is carried out to stop the layer from working or to disrupt the layer's performance.

**Table 3.** List of network attacks on WSN layers.

| References | Attacks | Layer | Routing-Based Attacks |
|---|---|---|---|
| [70,71] | Interception, radio interference, jamming, tempering, Sybil assault | Physical layer | Sybil assault |
| [64,72] | Replay attack, jamming assault, spoofing, altering routing assault, Sybil assault, traffic analysis, monitoring, exhaustion, collision | Data link layer | Routing assault, Sybil assault |
| [73,74] | Black hole assault, worm hole assault, sink hole assault, gray hole assault, selective forwarding assault, hello flood assault, misdirection assault, internet smurf assault, spoofing assault | Network layer | Black hole assault, worm hole assault, sink hole assault, gray hole assault, selective forwarding attack, hello flood assault |
| [75,76] | De-synchronization, transport layer flooding assault | Transport layer | ——— |
| [67,77] | Spoofing, path-based DoS, alter routing assault, false data ejection | Application layer | Alter routing assault |

Table 4 presents the features of specific attacks on WSN layers. The attack features show the damage executed upon each layer when attacked by a particular attacker. The attacks carried out on WSN layers are categorized into two types: internal and external. Most external attacks capture sensitive information by using malware such as worms, Trojan horse viruses, phishing, and other methods to obtain access to government and corporate websites, applications, and security systems. Network employees with connections to servers and classified documents who are dissatisfied are more likely to target and rob intellectual property. An insider threat occurs when a current or former employee, consultant, or business partner obtains access to an organization's network, system, or data and intentionally misuses them or when such an individual's access results in misuse.

Table 4. Features and types of attacks related to WSN layers [78–80].

| Attack | Layer | Type | Features of Attack |
|---|---|---|---|
| Eavesdropping | Physical | Ex | Without the node's awareness, it overhears and intercepts data in its transmit coverage area. |
| Basic jammers | Physical | Ex | Intentional radio emissions obstruct or discourage data transmission. |
| Intelligent jamming | Data link | Ex | Since the protocol rules for data delivery are defined, data packets are explicitly targeted. Collisions with adjacent nodes can occur if the filled radio channels are used. |
| Collision | Data link | In | Collisions with adjacent nodes can occur if the filled radio channels are used. |
| Replay attack | Network | In | Repetition of a successful data transfer. |
| Black hole | Network | In | Failing to forward all submitted data packets, including its own. |
| Sink hole | Network | In | Fake information is advertised to construct a point of interest for other nodes. |
| Sybil assault | Network | In | Using the network to present different identities. |
| Node replication | Network | Ex/in | Physically grabbing a node, replicating it, and redeploying it into the network. |
| Open worm hole assault | Network | In | The attacker gains access to the source and destination, builds a bogus path without the users' awareness, monitors the information, and sends it to the destination. |
| Data integrity | Transport | In | During delivery, data is compromised by the attacker modifying the content or inserting fake messages. |
| Energy drain | Transport | Ex/in | Sends as many link institution requests as possible to a specific node or nodes. |
| Exhaustion | Data link | In | Energy resources are wasted, causing the target node to conduct unneeded calculations or to receive or deliver data. |
| Tampering | Physical | In | Retrieves cryptographic material such as cipher keys. |
| Hello flood attack | network | Ex/in | Sends a "hello" packet to a neighboring node and modifies its network topology. |
| Attack on reliability | Application | Ex/in | Places the node in the communication line to produce false data or questions. |
| Malicious code assault | Application | Ex/in | Injects a "bug" into the program that causes it to crash or assumes full charge of the application's resources. |
| DoS | Multi-layer | Ex/in | A broad assault that could involve many other attacks occurring at the same time. |
| Man-in-the-middle assault | Multi-layer | Ex/in | Sniffs the network to intercept, without the network's knowledge, contact between two sensor nodes at the key exchange stage. |

## 10. Network Security Attacks and Issues of Data Link Layer Protocols

Table 5 describes the network security attacks on popular data link layer protocols and states the network consequences. We discuss the effects of network security attacks on network performance. Concerns regarding network ramifications for major data link layer issues, such as flow control, quality of service (QoS), and jamming assaults, are commonly articulated. At the data link layer, flow management is a design problem. It is a technique that monitors the correct flow of data from the sender to the recipient. The sender must relay data or information at a high rate, allowing the receiver to grasp and process the information. Flow control in the data link layer essentially limits and coordinates the number of frames or data a sender can send before waiting for a response from the recipient [81]. QoS applies to traffic prioritization and resource reservation management

systems that help network infrastructure handle packet loss, latency, and jitter. It can give various programs, customers, or data flows different goals or promise a certain degree of output for a data flow; QoS is especially critical when transporting traffic with special needs. Jamming attacks can be considered a subset of DDoS (distributed denial of service) [82] attacks that disrupt the radio channel by delivering many short packets to overburden the system. A DDoS assault on a WSN modifies the routing protocol information using the Dynamic Source Routing protocol (DSR), resulting in a massive quantity of unwanted traffic and the network or website being inaccessible [60,80]. In this way, traffic flow is highly affected due to high latency. WSN efficiency will likely be disrupted if an attacker uses a vital jamming source. Thus, it can be concluded that WSNs face numerous restrictions, such as a low computing capacity, insufficient memory and energy resources, physical capture vulnerability, and the use of unreliable wireless communication networks [83].

**Table 5.** Discussion of network security attacks on data link layer protocols and possible solutions in WSN.

| References | Data Link Layer Protocols | Attack | Issue Category | Network Consequence | Solution |
|---|---|---|---|---|---|
| [84] | HDLC (High-Level Data Link Protocol) | Hidden node attack DoS attack | Flow control QoS Jamming attack | A malicious node is added, or a regular node is injected with malicious code or a request. The node can infect the whole network, or network properties can be forged, causing harm to network integrity with denial of service. | For hidden node attacks, applying RTS and CTS mechanisms can help prevent hidden node attacks. These mechanisms validate data sending and receiving for reliable flow control and data connection [85]. |
| [86] | Point-to-Point Protocol | Switch spoofing CAM Table Exhaustion attack | QoS Jamming attack | The intruder sets up a device to impersonate a switch and sends DTP negotiation frames. Both VLANs borne on the trunk are sent to or collected by the attacker. The attacking host can then access traffic from several VLANs. CAM overflow allows an attacker to listen in on a conversation and conduct man-in-the-middle attacks. | Applying port-security protocols and dynamic MAC restrictions can help secure switch spoofing and CAM table exhaustion attacks [87]. |
| [88] | CDP (Cisco Discovery Protocol) | Fake access points Flooding attack | Flow control QoS Jamming attack | An evil twin attack that provides the attacker with direct access into the network, causing harm or loss of confidential information in a fake access point attack. Flooding attacks are quite similar to DoS attacks. The network is bombarded with a large number of requests or false inputs, resulting in a flood of streamed inputs. | A fake access point is nearly impossible for Wi-Fi devices, so using a VPN can encapsulate the Wi-Fi session in another layer of encryption. Moreover, wireless intrusion prevention systems (WIPS) can detect the presence of fake access points. The onsite firewall should be configured or an intrusion prevention system (IPS) should be installed to detect anomalous traffic patterns [89]. |
| [90] | Stop-and-Wait Protocol | DoS attack SYN flooding | QoS Jamming attack | In a DoS assault, the attacker attempts to disturb the services of a host connecting to the Internet by overwhelming the intended computer or resource with unnecessary requests, rendering computer or network resources unavailable to the engaged users. A SYN flood is a DDoS attack that causes the server to be inaccessible to legitimate traffic. | Installing network rate-limiting devices; installing business apps to gain network insight and to observe and analyze traffic from multiple regions [91]; and installing an intrusion prevention system (IPS) to detect odd traffic patterns can prevent attacks. |
| [92] | ARP (Address Resolution Protocol) | Session hijacking Fake access points | Flow control QoS Jamming attack | Session hijacking, also known as TCP session hijacking, operates by stealthily obtaining the session ID and posing as the authorized user. A fake access point attack is an evil twin attack in which fake access points appear just like actual ones and deceive users. | Personal VPN solution software can encrypt all data, not just traffic to the webserver. End-to-end encryption between the user's browser and the web server using encrypted HTTP or SSL prevents unwanted session ID access. Session ID detectors may also be used to examine such issues [93]. |

**Table 5.** *Cont.*

| References | Data Link Layer Protocols | Attack | Issue Category | Network Consequence | Solution |
|---|---|---|---|---|---|
| [94] | Spanning Tree Protocol | STP manipulation attack | Flow control QoS | STP prevents bridging loops in a redundant switched network system. The attacker spoofs the topology's root bridge, causing STP to be recalculated; the attacker broadcasts an STP configuration/topology change BPDU. | Root guard and BPDU guard assist against STP manipulation attacks by ensuring that no user data is transmitted over a port that is in the root-inconsistent state [95]. |
| [85] | LLDP (Link Layer Discovery Protocol) | Spoofing attack Selective forwarding attack | Flow control QoS | Spoofing may be used to gain access to a target's personal information and disseminate ransomware via infected links or attachments aimed at stealing information or distributing malware. In a selective forwarding assault, the attacker loses certain packets at an arbitrary time, which can be used to protect against an insider packet drop attack. | Spoofing attacks can be avoided by spoofing detection tools, which improve the ability to identify and stop them before they can do any damage. Packet filtering can filter out and block packets, which can avoid IP spoofing [96,97]. To dissuade selective forwarding-based DoS attacks proactively, a defense method for detection and avoidance, such as a preventative routing algorithm, must be implemented [98]. |
| [99] | Ethernet protocol | MAC flooding Port stealing | Flow control QoS | MAC flooding is a cyberattack that jeopardizes the security of network switches. The hacker uses this technique to intercept sensitive data being transported over the network. Port stealing is a form of assault in which someone "steals" traffic from one Ethernet transfer port to another. This type of attack causes anyone to accept packets intended for a particular device. | Because the switch's MAC address table contains incorrect MAC address and port mappings, the port security function can protect it from MAC flooding assaults [100]. In insecure situations, enterprise-grade switches may be used to protect the environment [101]. |
| [102] | Sliding window protocol | SYN flooding DoS attack | Flow control QoS Jamming attack | An SYN flood is a DDoS attack that consumes all available server resources by overloading all open ports on a targeted server.A denial-of-service attack seeks to interrupt the services of a host connected to the Internet by flooding the intended computer or resource with unnecessary requests, rendering the computer or network resources unavailable to their engaged users. | Installing network rate-limiting devices is recommended. It is a good idea to install an intrusion prevention system (IPS) to identify abnormal traffic patterns in order to avoid SYN flooding assaults, configure the onsite firewall for SYN assault thresholds, and implement SYN flood defense [91]. Preparing a DoS attack response plan and protecting the infrastructure, investigating black hole routing, upgrading firewalls and routers to reject fake traffic, and applying the latest security updates to routers and firewalls can prevent attacks [103]. |

## 11. Existing Literature in WSN Domain

Table 6 presents the recent work undertaken within the WSN domain along with problems addressed and their contributions. Significant research gaps in the stated research are also detailed and mapped with respect to major data link layer issues such as QoS, flow control, security, and implementation. The research gaps mentioned in the table highlight gray areas where future researchers may improve, enhance, and present more efficient and effective solutions.

**Table 6.** Contributions and research gaps of related recent work undertaken.

| Sr.# | Authors Name and Year | Problem | Contribution | Research Gaps |
|------|----------------------|---------|--------------|---------------|
| 1. | Thiago C. et al., 2020 [104] | There is no significant variation in quality measurement in insensitive systems using scalar and visual sensors to perform control functions. | Failures in hardware, networking, and vision coverage are considered when assessing the efficiency of wireless sensor networks. | Lacks reliability involving monitoring modifications to the system due to faults or repairs. Self-diagnosis features should be a part of enhancing QoS. |
| 2. | M. Faheem et al., 2018 [105] | A significant issue is noted when sensor nodes forward a sensor node that closes the static sink. Massive amounts of data from more distant sensors are being collected in the deployed network. Thus, the network's multi-to-one traffic pattern stems from these sensors bearing a large traffic load. The network partition problem is caused by vulnerability to energy depletion when consuming energy. | SIRP (self-optimized intelligent routing protocol) is built on bio-inspired principles for WSN-based SG applications. | The communication architecture that ensures diverse QoS-aware data should have been considered for collection with minimal data replication for multiple WSN-based SG applications. This would have improved flow control and QoS. |
| 3. | Arslan et al., 2020 [106] | IoT-based WSNs face obstacles due to various environmental changes. The variety of sensor nodes required to monitor vast areas is increasing. | To reliably share sensor data, the NRF protocol can be used. Camera processing and sensors to capture illumination, moisture, humidity, temperature, and other parameters are used to detect weeds. | The framework does not meet modern requirements as it lacks mobile application control of the robot, which could have been a positive asset toward achieving better QoS for users. |
| 4. | Chi-Tung Chen, Cheng-Chi Lee, Iuon-Chang Lin, 2020 [107] | Time synchronization is another crucial and challenging topic for WSNs. The machine must provide a suitable logical time clock for all devices and objects in IoT environments. Any attacker or malicious node in an IoT system can attempt to disrupt clock synchronization. | Prevention capabilities are quantitatively superior, and authentication efficiency in the IoT can be improved qualitatively. High performance, low computing and connectivity costs, and the lower consumption of resources were positive attributes. | The system can cause hazards to emerge in heterogeneous IoT settings. Different heterogeneous IoT implementations may cause severe network security difficulties. This represents a threat to QoS and to the prevention of jamming attacks. |
| 5. | Patricia A. et al., 2019 [108] | Spatiotemporal resolution constraints are typical, leading to issues with traditional air quality control systems, such as system non-scalability or reduced personal exposure data storage. | The results show that the method for discriminating and quantifying volatile organic compound concentrations is efficient. | The system lacks the implementation of various sensor nodes to check actual conditions and configure sensors in the region. The system is limited to research only. |
| 6. | Xiaomin Li et al., 2020 [109] | Agricultural WSNs face many difficulties, such as multitasking with critical problems of data collection and processing to maintain data accuracy and reduce lag for better performance. | A double selection approach determines the right node and sensor network that satisfies data quality and collection time constraints. A data collection algorithm is developed based on a set of data quality values. | The mentioned algorithm has no capacity for data collection from the natural environment, limiting its use to research only. |
| 7. | Nalluri Prophess Raj Kumar and Josemin Bala Gnanadhas [110] | The network of wireless sensors is incredibly resource-restricted. Because sensor nodes are battery-powered and deployed in hazardous areas, it is difficult to recharge or replace batteries after deployment. Stable routing protocols are needed to improve network life and offset energy consumption. | The algorithm outperforms traditional protocols in terms of efficiency parameters such as network energy consumption, average sensor node energy consumption, packet failure percentage, packet distribution ratio, and network throughput. | The route is not configured in the framework, which defies the route policy that triggers packet loss if the base station is far away; the ZH must waste most of its energy on data transfer, causing flow control and QOS issues. |
| 8. | Khalid Haseeb et al., 2020 [111] | In terms of generation, energy, transmission, and memory capacities, sensors have limited resources that can adversely affect agricultural production. In addition to their performance, the security and protection of these IoT-based agricultural sensors are critical for malevolent attackers. | The device has dramatically improved communication performance, network throughput, packet drop ratio, network latency, energy consumption, and routing overhead for intelligent agriculture. | The framework lacks the evaluation required to match the consistency and performance of the device in a mobile-based IoT environment that cannot be configured; thus, it does not meet the modern requirements of IoT frameworks. |
| 9. | Khashan, O et al., 2021 [112] | Despite developing a novel lightweight cryptographic approach for WSNs, there are various limitations, including flexibility, authentication power resource management, and critical management processes. | A lightweight cryptographic technique, "FlexCrypt", is developed to address the existing issues. | There exists a need to resist more attacks rather than considering fewer. |

## 12. Security Issues concerning the Data Link Layer

As WSNs are broadcast in nature, several vulnerabilities might arise as a result of attacks that can easily disrupt, modify, or insert malicious data.

There are several types of attacks that affect the data link layers and change the flow of traffic. At the data link layer, flow management is a strategy for ensuring that data flows correctly from the source to the recipient and is an issue of design. The transmitter must

transfer data or information at a rapid rate so that the recipient can comprehend and process it. Flow control in the data link layer effectively limits and controls the number of frames or data that a sender may transmit before waiting for a response from the recipient [81]. In such scenarios, DoS assaults that consume resources and potentially squander network bandwidth are widespread and well known.

Furthermore, a node capture attack can capture a network node to get its cryptographic keys and protocol status and then redeploy malicious nodes across the network [15]. Similarly, the classical MAC protocol, which is based on a competitive process, is incapable of meeting the requirements of WSNs. Because content-based MAC protocols have several handshakes, there is a significant possibility of data collisions that cause energy waste in WSNs. As a result, to protect node energy, MAC protocols for WSNs often use pre-planned techniques such as TDMA [113].

Identifying critical QoS criteria to evaluate network performance is part of success assessment. The QoS parameters in network performance indicate that these factors substantially affect network metrics in communications, with the impact differing based on the network criteria and network communication aspects [114]. The QoS parameters include network performance analysis, availability, bandwidth, throughput, transit delay, jitter, resilience, protection, and packet loss rate [81]. QoS can assign distinct goals to different programs, clients, or data flows or promise a set production level for a data flow; thus, it is critical when conveying traffic with unique demands [82]. Implementing QoS can help organizations generate significant profits. Performance improvements in applying QoS can help reduce packet loss, error, and latency by defining and prioritizing sensitive applications based on their network traffic. Because QoS network traffic recognition prioritizes traffic and creates application-specific policies, costly and high-performance network bandwidth can be used mainly or exclusively by the applications that require it, resulting in improved network utilization. An entity can more effectively direct network traffic to its destination by defining the application connected to a given network link and by adding application-specific policies, which optimizes traffic routing. QoS can help minimize network congestion by dropping or throttling low-priority traffic during high-use times and via application-specific traffic filtering to relieve congestion in important network sectors. Table 7 describes the QoS parameters, their necessity for enhancing network performance, and details how their absence affects network performance [115–117].

**Table 7.** QoS parameters and their effects on network performance at the data link layer.

| Parameters | Network Performance |
| --- | --- |
| Network availability | The availability of a network will influence QoS and network performance. The user or software may receive unexpected or unwanted results if the network is down. The availability of several things used to build a network, such as redundant network devices or interfaces; power supplies in routers and switches; processor cards; resilient networking protocols; various physical links; etc., can render the network unavailable for users, causing a decline in network performance. |
| Bandwidth | Network carriers have a limited amount of bandwidth; when oversubscribing to bandwidth, a customer must always have it available. This encourages consumers to bid for the limited amount of B.W. available. They receive B.W. depending on the traffic generated by other network users at any given time. When subscribers use the same network infrastructure, guaranteed B.W. subscribers must have preference over available B.W. subscribers' traffic to ensure B.W. subscribers' SLAs are reached even when the network is congested. |
| Throughput | Throughput is the number of packets that successfully reach their destinations. Bits per second or data per second may describe the throughput power. The arrival of packets is critical to the high-performance operation of a network. When using services or apps, people expect their requests to be heard and responded to quickly. Low throughput implies packet loss, which contributes to bad or slow network efficiency. Throughput is a metric that calculates network speed, but a low value may impact network efficiency, causing packet loss, latency, and jitter. |

**Table 7.** *Cont.*

| Parameters | Network Performance |
| --- | --- |
| Transit delay | The time it takes an application to travel from the ingress (entry) point to a network's egress (exit) point is referred to as network latency. Delay can cause severe QoS problems with applications such as video conferencing and fax delivery, which time out and terminate because of an unreasonable delay. Network propagation delay can cause ingress queuing delays for traffic entering a network node, traffic conflict at each network node, and egress queuing delays for traffic leaving a network node. At each network hop, data is distributed over the physical network medium. |
| Jitter | The difference in delay reported by different packets in the same traffic flow is known as jitter, as is high-frequency delay variance. The most crucial problem for QoS is jitter caused by variations in queue wait times for consecutive packets in a flow. Jitter is not tolerated by some forms of traffic, especially real-time traffic such as video conferencing. Jitter is present in all transportation networks. Jitter thresholds have little effect on service quality if they are below the specified tolerance level. |
| Resilience | Quality of service (QoS) is a critical consideration in the architecture of IP-based multimedia and multiservice networks. Network resilience refers to a network's ability to survive network assaults and poor results. If the network becomes more vulnerable to attacks, such as sniffing, spoofing, and malicious operations, data confidentiality will be compromised, and data loss will occur. Every well-designed recovery plan must account for the multiple reliability needs of individual traffic flows to prevent unnecessary bandwidth consumption for standby links and decide which flows to defend against network failures as well as the degree to which they must be defended. |
| Loss | If a network node gets overburdened, it may lose packets and fail. TCP (Transmission Control Protocol) is a networking protocol that defends against packet loss by retransmitting packets lost by the network. As network congestion increases, more packets are lost, resulting in increased TCP transmission. Since most B.W. is used to retransmit lost packets, network capacity can deteriorate if congestion persists. |

## 13. Conclusions

WSNs and the IoT have been rapidly and enormously evolving. These technologies have revolutionized many aspects of life and can be observed throughout the world, independently or in collaboration. A WSN is a wireless sensor cluster that communicates and functions together. Industrial connectivity technologies have already been advancing for many years to help meet evolving needs in traditional application scenarios, such as factory automation and distributed process control systems, or to cope with continuous demands for improved efficiency. The Fourth Industrial Revolution and the IIoT plainly illustrate that this evolutionary shift must be accelerated and expanded to encompass growing research disciplines and severe technical problems, since neither new demanding communication needs nor creative applications can be fulfilled while solely depending on the assistance provided by today's communication technologies. This article covered all the aspects of WSN from its architecture and security requirements to its applications and implementations as well as the issues and challenges it faces. Such information can help to clarify a research perspective on WSN in IoT systems. Because little work has been undertaken on the data link layer and its relationship to WSN infrastructure, and due to existing major data link layer issues, its use has not been adequately supported or considered, resulting in a lack of innovative and efficient solutions within WSN applications. There is a significant research gap in the development of solutions to data link layer issues, including QoS, security, and flow control, which requires the immediate attention of researchers to develop modern, effective, and efficient solutions that support the issues, challenges, and limitations of the data link layer protocol. Current active security algorithms; QoS and flow control improvement techniques; and efficient implementation and deployment schemes for the domain of the data link layer in WSN frameworks represent future goals. Future work should address the problems of WSN area coverage and develop efficient solutions to existing WSN issues and challenges. Broadening the implementation and application of WSN in the IR4.0 and IIoT is a significant goal required to classify the issues and challenges facing the IR4.0 and provide effective solutions.

**Author Contributions:** M.Z.H.—main author of this review article who search and analyse the research idea; Z.M.H.—the supervisor who moderated & approved the review as well as provide the funding for the project. All authors have read and agreed to the published version of the manuscript.

**Funding:** This work was supported by Geran Putra Berimpak Universiti Putra Malaysia, Vote Number 9659400. Our sincere thanks to Geran Putra Berimpak Universiti Putra Malaysia for their support.

**Data Availability Statement:** Not applicable.

**Conflicts of Interest:** The authors declare no conflict of interest.

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
