# Peer review of "Efficient and Secured Mechanisms for Data Link in IoT WSNs: A Literature Review"

_electronics, doi:10.3390/electronics12020458_

Round 1
Reviewer 1 Report
This paper covers all the aspects of WSN, from its architecture to security requirements to application and implementations to issues and challenges of WSN. This paper can help understand the research perspective of WSN in IoT systems. As minimal work was done to the data link layer and its relation to WSN infrastructure and major data link layer issues, it has not been adequately supported or considered causing the absence of innovative and efficient solutions within WSN applications. There is a significant research gap in the development of data link layer issues, including QoS, security, and flow control, that needs immediate attention of researchers to develop modern, effective, and efficient solutions that support these issues, challenges, and limitations data link layer protocol. The following comments and suggestions need to be considered:
1- The first two figures require redrawing for more custom illustration and the second one has wrong figure number in the caption.
2- It's better to merge the second section with the previous or succeeding sections, a sentence at its end (lines 73-74) need to be revised "Each of these concepts is defined in detail further down the page".
3- Figure 3 drawn with three dimensions, while only two dimensions are used. The reviewer do not know why it's black and white. Another sentence in (lines 90-91) "WSN's five tiers are detailed below 90 in Fig. [3]." requires revision in accordance with the caption of Figure 3.
4- Lines 119 - 128 need to be revised grammatically, moreover, adding a clear description for Figure 4 is mandatory.
5- Figure 4 require redrawing, it has the same reference with Figure 3.
6- Figure 7 require drawing with colors for more custom illustration.
7- In Table 5, the network consequence and solution columns need to be summarized for each reference under consideration.
8- Conclusion section is necessary. What are the new conclusions drawn from this review paper ?
9- In the text of the paper, reference 44 is cited after reference 29, reference 87 has green color.
Author Response
- The first two figures require redrawing for more custom illustration and the second one has wrong figure number in the caption.
Reply: Changes incorporated
- It's better to merge the second section with the previous or succeeding sections, a sentence at its end (lines 73-74) need to be revised "Each of these concepts is defined in detail further down the page".
Reply: Ambiguous sentence is removed, Changes incorporated
3- Figure 3 drawn with three dimensions, while only two dimensions are used. The reviewer do not know why it's black and white. Another sentence in (lines 90-91) "WSN's five tiers are detailed below 90 in Fig. [3]." requires revision in accordance with the caption of Figure 3.
Reply: Changes incorporated
4- Lines 119 - 128 need to be revised grammatically, moreover, adding a clear description for Figure 4 is mandatory.
Reply: Changes incorporated
5- Figure 4 require redrawing, it has the same reference with Figure 3.
Reply: Changes incorporated
6- Figure 7 require drawing with colors for more custom illustration.
Reply: Changes incorporated
7- In Table 5, the network consequence and solution columns need to be summarized for each reference under consideration.
Reply: Changes incorporated
8- Conclusion section is necessary. What are the new conclusions drawn from this review paper ?
Reply: Changes incorporated
9- In the text of the paper, reference 44 is cited after reference 29, reference 87 has green color.
Reply: Changes incorporated
Reviewer 2 Report
The paper introduced typical security protocols in the Internet of Things (IoT). The architecture of IoT in terms of energy consumption, mobility, information transmission, QoS, and security was presented. Although many existing solutions had been introduced, there are still some issues to addressed.
1. In Section 5 RESEARCH ISSUES OF WSN IN IOT, the authors mentioned: Apps' potential to interact with sensors, other users, and the cloud is called "coverage and connectivity." Then the authors listed some related research topics including "localization techniques". In this place, it is strongly recommended that the authors mention some related work about both connectivity and localization, such as the article titled as Connectivity based DV-hop localization for Internet of Things, which was published in IEEE transactions on vehicular technology. It is suggested that the authors refer to this kind of recent related work.
2. The tables in the paper should be modified. For example, in Table 1, there are too much words in the field "Required Research development". It is strongly recommended that the authors use fewer words to summarize the requirement.
3. The English writing should be well improved. It is hard to understand the sentences in the paper. For example, in the abstract, the following sentence is too long. It is very hard to understand its meaning. "The critical principle of IoT consists of making existing companies sufficiently intelligent to understand the need for significant fault mitigation and short-cycle adaptation to enhance effectiveness to increase financial profits."
Author Response
- In Section 5 RESEARCH ISSUES OF WSN IN IOT, the authors mentioned: Apps' potential to interact with sensors, other users, and the cloud is called "coverage and connectivity." Then the authors listed some related research topics including "localization techniques". In this place, it is strongly recommended that the authors mention some related work about both connectivity and localization, such as the article titled as Connectivity based DV-hop localization for Internet of Things, which was published in IEEE transactions on vehicular technology. It is suggested that the authors refer to this kind of recent related work.
- The tables in the paper should be modified. For example, in Table 1, there are too much words in the field "Required Research development". It is strongly recommended that the authors use fewer words to summarize the requirement.
Reply: Changes incorporated
- The English writing should be well improved. It is hard to understand the sentences in the paper. For example, in the abstract, the following sentence is too long. It is very hard to understand its meaning. "The critical principle of IoT consists of making existing companies sufficiently intelligent to understand the need for significant fault mitigation and short-cycle adaptation to enhance effectiveness to increase financial profits."
Reply: Changes incorporated
Round 2
Reviewer 1 Report
The authors addressed my corrections.